# Investigation of an Output Voltage Harmonic Suppression Strategy of a Power Quality Control Device for the High-End Manufacturing Industry

**DOI:** 10.3390/mi13101646

**Published:** 2022-09-30

**Authors:** Chengkuan Wan, Kai Li, Lin Xu, Chao Xiong, Lingang Wang, Hao Tang

**Affiliations:** 1School of Automation Engineering, University of Electronic Science and Technology of China, Chengdu 610056, China; 2Kuntengtaike (Chengdu) Technology Co., Ltd., Chengdu 610300, China; 3State Grid Sichuan Electric Power Co., Ltd., Chengdu 610041, China

**Keywords:** harmonic suppression, high-end manufacturing, voltage feedback, current feed-forward, compound control strategy

## Abstract

Due to the influence of nonlinear loads, power quality control devices under the traditional double closed-loop control strategy suffer from a large output voltage harmonic distortion rate and a slow harmonic suppression response and cannot meet the high-quality power supply requirements of high-end manufacturing. A compound control strategy based on voltage feedback and current feed-forward is proposed to solve the voltage quality problem under a nonlinear load. Firstly, based on the mathematical model of power quality control device, the working principle and voltage current coupling relationship are analyzed. Then, an output voltage compound control strategy based on feed-forward and feedback is proposed, and the harmonic suppression mechanisms are deduced and analyzed. Finally, simulation and experimental results are presented to show that, compared with the traditional double closed-loop control strategy, the harmonic suppression effect of the composite control strategy proposed in this paper can be increased by 2.2%, and the response time is decreased to 100 ms.

## 1. Introduction

With the upgrading of China’s manufacturing industry, a number of high-tech industries represented by precision processing, chip and electronic manufacturing, data centers and so on have developed rapidly. Considering that the high-precision production equipment in these high-tech industries are very sensitive to voltage sag, three-phase imbalance and harmonic voltage [1,2], the power quality problem becomes more and more prominent, so ensuring high-quality power supply for loads has become one of the problems that needs to be solved urgently [3,4,5].

Based on modern power electronics technology and modern control theory, the cascaded power quality improvement device [6,7] has been developed to solve the full power quality problems, such as three-phase unbalance, voltage harmonic, and voltage sag. The cascaded power quality improvement device [6,7] consists of a phase-shifting transformer, rectifier units, H-bridge inverter units and energy storage battery packs. The phase-shifting transformer [8] converts AC to AC with different phases in N groups (N is the number of the transformer windings); N groups of AC, respectively, convert into DC after passing through multiple rectifier units, and the DC bus connected with the energy storage battery pack can ensure that the energy storage battery continues to supply power to the loads after temporary power loss on the AC side, so as to solve the problem of voltage sag caused by voltage drop on the AC side effectively; After the parallel of the N group energy storage batteries and rectifier units, they are connected with the H-bridge inverter units, respectively, in which the outlet of each inverter unit is connected to the output side in cascade [9]. In this way, the power grid is isolated from the user, so the loads can obtain high-quality and reliable power, which can meet the working needs of high-sophisticated equipment.

Affected by nonlinear loads, unbalanced loads, semiconductor devices, line impedance and other factors, the voltage at the common connection point (PCC) on the output side of the cascaded power quality improvement device will be distorted [10]. Therefore, the cascaded power quality improvement device needs to have an excellent voltage harmonic improvement function, especially to ensure power quality under a nonlinear load [11]. At present, many control strategies have emerged to optimize the distortion of an output side voltage waveform caused by a hybrid load [12,13,14]. The study in [15] combined improved repetitive control with instantaneous control, and introduced repetitive control to eliminate periodic harmonic disturbances, but the dynamic response was insufficient. The study in [16] converted the 5th and 7th harmonics and negative sequence components into DC through multiple coordinate transformations for control, but the use of a large number of low-pass filters reduced the dynamic performance of the system. In the case of nonlinear load dynamic change, the PI controller cannot achieve a good control effect and a fast response to harmonics. Based on the proportional resonant controller, the study in [17] and the study in [18] adopted a feed-forward control strategy and introduced additional impedance, which offset the disturbance of part of the harmonic current and suppressed the harmonics to a certain extent. However, feed-forward compensation needs to compensate each harmonic one by one. When designing the controller, it is necessary to design each harmonic controller, which increases the design difficulty of the controller and the computational burden of the processor. The study in [19] proposed an equivalent impedance reshaping strategy to reduce the output voltage THD, and the effect was remarkable, but did not give the selection method of virtual harmonic impedance and did not analyze the stability of the system. The studies in [20,21] introduced droop control to reduce the THD of the output voltage. However, inverters with different impedance types need to select different droop control equations, which lack adaptability and flexibility.

Considering that the traditional double closed loop control algorithm is difficult to improve the voltage harmonics at the common connection point, a composite control strategy based on voltage feedback and current feed-forward is proposed. By detecting the PCC point voltage for harmonic voltage feedback control and detecting the PCC point current for harmonic current feed-forward control, the compound control strategy can not only improve the output voltage harmonic compensation effect but can also ensure the output voltage harmonic compensation speed, so as to be suitable for suppressing voltage harmonic distortion caused by the rapid change of hybrid load. In order to verify the effectiveness of the proposed composite control strategy, a simulation and experimental verification were carried out through the simulation model and test platform and it was proven that the composite control strategy based on feed-forward and feedback can effectively improve the harmonic distortion rate. The contribution of this paper is listed as follows:

1. A novel composite control strategy based on voltage feedback and current feed-forward is proposed, which can compensate not only stable nonlinear loads but also rapidly changing nonlinear loads;

2. The harmonic suppression mechanism and realization method of voltage feedback and current feed-forward in the proposed compound control strategy are studied in depth;

3. The proposed composite control strategy is simulated and tested, and the compensation effect and dynamic response performance of the control strategy are verified.

## 2. System Analysis

### 2.1. Introduction of a Cascaded Power Quality Improvement Device

The topological structure of the cascaded power quality improvement device is shown in Figure 1 [22]. The device is connected in series in the distribution network system, which can ensure that the power quality of the grid and the load side are isolated from each other.

In Figure 1, the device is composed of a phase-shifting transformer, a bypass switch cabinet and a cascade power unit. The output voltage stability can be ensured by the cascaded power quality improvement device when the power quality problems occur on the grid side. The phase-shifting transformer can not only achieve electrical isolation and voltage level conversion but can also eliminate the harmonic influence of the uncontrolled rectifier part of the device on the power grid. The secondary windings of the phase-shift transformer are connected to the cascaded power unit modules $A_{1},A_{2},\cdots ,A_{n}$, $B_{1},B_{2},\cdots ,B_{n}$ and $C_{1},C_{2},\cdots ,C_{n}$. N cascaded power units are connected in series in phase A, thereby providing the 10 kV line voltage via superposition. Phase B and C are similar to phase A.

Figure 2 shows the new power unit which is the basic unit of the cascaded power quality improvement device. Three-phase uncontrolled rectifier bridge, DC bus link, H bridge inverter unit, energy storage unit and related control and protection components are the main components of the power unit. The characteristic of the power unit is that when the power grid voltage sag fault occurs, the new power unit can realize zero delay power supply to the DC bus through the power diode. When the power grid voltage is normal, the power unit can actively charge and discharge the battery pack at low power, so as to optimize the service life of battery pack and save the cost and volume of the power unit.

### 2.2. Principle Analysis

Figure 3 shows the topology of an equivalent circuit of the cascaded power quality improvement device. In Figure 3, *u_a_*, *u_b_*, *u_c_* represent the output voltages of the device, *i_a_*, *i_b_*, *i_c_* represent the output currents of the device, *u_pa_*, *u_pb_*, *u_pc_* represent the three-phase voltages of the PCC, and *i_La_*, *i_Lb_*, *i_Lc_* represent the load currents.

According to the circuit topology shown in Figure 3, the expressions between output voltage, current and PCC point voltage can be obtained:(1)$$\left\{ \begin{array}{l} u_{a}=\left(L+L_{f}\right)\frac{di_{a}}{dt}+Ri_{a}+u_{pa}\\ u_{b}=\left(L+L_{f}\right)\frac{di_{b}}{dt}+Ri_{b}+u_{pb}\\ u_{c}=\left(L+L_{f}\right)\frac{di_{c}}{dt}+Ri_{c}+u_{pc} \end{array}\right.$$
where *R* represents the sum of the power switch tube loss resistance and the line loss resistance; *L_f_* and *C_f_* are the resistance values of the three-phase filter inductor and filter capacitor; and L represents the sum of the line inductance on the output side of the device and the leakage inductance of the transformer.

Three-phase time-varying ac is not conducive to the design of the control system. This paper uses coordinate transformation to obtain a mathematical model based on d and q coordinates and convert the fundamental sine variables into DC variables. After converting the time-varying AC quantity to the DC quantity, the difficulty of the control system design can be reduced and the static error-free control can be realized. After the expression (1) is transformed by Park, the following expression can be obtained [23]:(2)$$\left\{ \begin{array}{l} u_{d}=\left(L+L_{f}\right)\frac{di_{d}}{dt}+Ri_{d}-\omega \left(L+L_{f}\right)i_{q}+u_{pd}\\ u_{q}=\left(L+L_{f}\right)\frac{di_{q}}{dt}+Ri_{q}+\omega \left(L+L_{f}\right)i_{d}+u_{pq} \end{array}\right.$$

## 3. Compound Control Strategy under Voltage Feedback + Current Feed-Forward

### 3.1. Control Principle

The compound control strategy based on feedback and feed-forward proposed in this paper is to add a harmonic voltage feedback control loop and a harmonic current feed-forward control loop, which is on the basis of the fundamental wave voltage and current double closed-loop control. The control principle block diagram is shown in Figure 4.

In Figure 4, voltage feedback control ensures the precision of harmonic voltage control, and current feed-forward control ensures the dynamic response speed of harmonic voltage control. Voltage feedback control is when three voltage sensors detect the PCC point voltage, extract each harmonic voltage component to obtain the feedback voltage value and then superimpose it to the command voltage value for control. Current feedforward control is when three current sensors detect the current at the PCC point, and then extract the harmonic components and compensate the virtual impedance to suppress the harmonic components.

### 3.2. Analysis of the Voltage Feedback Control Strategy

The harmonic voltage feedback control strategy proposed in this paper is shown in Figure 5. In the figure, the harmonic voltage feedback control realizes the suppression of the harmonic voltage at the PCC point.

The harmonic voltage feedback control loop proposed in this paper is mainly for rectifier bridge-type nonlinear loads, such as inverter air conditioners, energy-saving lamps and voltage-type inverters. The core of the control method is based on the fundamental wave control by adding the harmonic voltage loop control to achieve the purpose of 5, 7 and 11 harmonic suppression. Specifically, in order to achieve the purpose of 5th harmonic compensation, the first step is to transform the extracted 5th harmonic component into direct flow through −5ωt transformation, and then through low-pass filtering, the 5th harmonic compensation is achieved through PI control. The control principle of the remaining harmonics is the same. The theoretical derivation is as follows.

If the three-phase voltage at the PCC point contains only fundamental components, the expression can be expressed as:(3)$$\left\{ \begin{array}{l} u_{pab}=U\sin\left(\omega t\right)\\ u_{pbc}=U\sin\left(\omega t-2\pi /3\right)\\ u_{pac}=U\sin\left(\omega t+2\pi /3\right) \end{array}\right.$$

Then the 5th harmonic voltage can be expressed as:(4)$$\left\{ \begin{array}{l} u_{pa-5}=U_{5}\sin\left(5\omega t\right)\\ u_{pb-5}=U_{5}\sin\left(5\omega t-5\times 2\pi /3\right)=U_{5}\sin\left(5\omega t+2\pi /3\right)\\ u_{pc-5}=U_{5}\sin\left(5\omega t+5\times 2\pi /3\right)=U_{5}\sin\left(5\omega t-2\pi /3\right) \end{array}\right.$$

The 5th harmonic voltage is converted into a negative sequence, and the d-axis rotates clockwise at an angular velocity of 5*ω*. Under the condition of “equal amplitude” transformation, transform to the (d, q) coordinate system, the value of the 5th harmonic in the coordinate system can be calculated. The calculation process is shown in expression (5). The subscript N5 in Formula (5) represents five negative sequence transformations.
(5)$$\begin{array}{ll} \left[\begin{array}{l} u_{d5}\\ u_{q5}\\ u_{05} \end{array}\right] & =T_{N5}\left[\begin{array}{l} u_{pa-5}\\ u_{pb-5}\\ u_{pc-5} \end{array}\right]\\ \; & =\frac{2}{3}\cdot \left[\begin{array}{ccc} \sin5\omega t & \sin\left(5\omega t+\frac{2\pi }{3}\right) & \sin\left(5\omega t-\frac{2\pi }{3}\right)\\ -\cos5\omega t & -\cos\left(5\omega t+\frac{2\pi }{3}\right) & -\cos\left(5\omega t-\frac{2\pi }{3}\right)\\ \frac{1}{\sqrt{2}} & \frac{1}{\sqrt{2}} & \frac{1}{\sqrt{2}} \end{array}\right]\cdot \left[\begin{array}{c} U_{5}\sin5\omega t\\ U_{5}\sin\left(5\omega t+\frac{2\pi }{3}\right)\\ U_{5}\sin\left(5\omega t-\frac{2\pi }{3}\right) \end{array}\right]\\ \; & \;=\left[\begin{array}{c} U_{5}\\ 0\\ 0 \end{array}\right] \end{array}$$

In the expression (5), set the 5th harmonic command value to 0. After the control system is stabilized, the 5th harmonic voltage will tend to 0. Obviously, the given value of the system controller is relatively simple, and it is easy to implement without considering the initial phase angle, frequency and other issues.

In addition, when the 5th harmonic does not exist in the linear load, expression (5) is equal to zero. For the linear load, the 5th harmonic voltage feedback control loop will not play a role, so the control algorithm can run stably when the load is linear. The principle of elimination of the 7th harmonic is the same, except that the 7th harmonic uses the 7th positive sequence transformation.

### 3.3. Analysis of the Current Feed-Forward Control Strategy

When nonlinear loads such as diodes and rectifiers are connected to the back end of the device, the harmonic current generated by the nonlinear load will form harmonic voltage on the line impedance. The virtual impedance can reshape the output impedance of the device at each harmonic frequency, thereby effectively solving the problem of voltage harmonics generated by nonlinear loads.

This paper adopts a current feed-forward control strategy based on virtual impedance. First, the dq component (including interference) of each harmonic current is obtained by rotating transformation, and the interference is filtered through a low-pass filter (LPF), and then each harmonic current component is obtained. Finally, each harmonic of the voltage is compensated separately. Among them, the harmonic impedance of each order can be obtained by the corresponding harmonic current empirical formulas (6) and (7) [24].

The positive sequence virtual impedance as:(6)$$\left(\begin{array}{l} u_{virtual,\alpha ,hp}\\ u_{virtual,\beta ,hp} \end{array}\right)=\left(\begin{array}{cc} -R_{hp,vir} & -\omega_{hp}L_{hp,vir}\\ \omega_{hp}L_{hp,vir} & -R_{hp,vir} \end{array}\right)\left(\begin{array}{c} i_{o\alpha hp}\\ i_{o\beta hp} \end{array}\right)$$
where $u_{virtual,\alpha ,hp},u_{virtual,\beta ,hp}$ represent the α,β components of the positive sequence virtual impedance voltage drop; h represents the harmonic order; p represents the positive sequence; $i_{o\alpha hp},i_{o\beta hp}$ represent the α,β components of the positive sequence harmonic current of the PCC point current; $R_{hp,vir},L_{hp,vir}$ represent the virtual resistance and virtual inductance of the positive sequence harmonic current; $\omega_{hp}$ represents the angular frequency of the positive sequence harmonic.

The negative sequence virtual impedance as:(7)$$\left(\begin{array}{l} u_{virtual,\alpha ,hn}\\ u_{virtual,\beta ,hn} \end{array}\right)=\left(\begin{array}{cc} -R_{hn,vir} & \omega_{hn}L_{hn,vir}\\ -\omega_{hn}L_{hn,vir} & -R_{hn,vir} \end{array}\right)\left(\begin{array}{c} i_{o\alpha hn}\\ i_{o\beta hn} \end{array}\right)$$
where, $u_{virtual,\alpha ,hn},u_{virtual,\beta ,hn}$ represent the α,β component of the negative sequence virtual impedance voltage drop; h represents the harmonic order; n represents the negative sequence; $i_{o\alpha hn},i_{o\beta hn}$ represent the α,β components of the negative sequence harmonic current of the PCC point current; $R_{hn,vir},L_{hn,vir}$ represent The virtual resistance and virtual inductance of the negative sequence harmonic current; $\omega_{hn}$ represents the angular frequency of the negative sequence harmonic.

When the voltage drop of virtual harmonic inductor and virtual harmonic resistor cancel each other with the harmonic voltage drop of each harmonic current on the line impedance, the harmonic suppression function can be realized. The overall control block diagram is shown in Figure 6.

## 4. Simulation Validation

### 4.1. Simulation Model Construction

GB/T 24337-2009 standard stipulates that the total harmonic distortion rate of public distribution network must be ≤4%. In order to verify the effectiveness of the proposed control algorithm, the system simulation model and experimental platform as shown in Figure 7 were built based on the system control schematic diagram.

Table 1 shows the parameter settings of the system simulation model.

### 4.2. Comparative Analysis of Different Control Algorithms

Figure 8 shows the changes in the harmonic distortion rate of A, B, C three-phase voltage under the voltage feedback control strategy. Figure 9 shows the changes of voltage harmonic distortion ratios of A, B, C three-phase voltages under the current feed-forward control strategy, and Figure 10 shows the changes of A, B, C three-phase voltage harmonic distortion ratios under the feedback and feed-forward composite control strategies. The starting time of the control strategy was at t = 0.2 s.

In Figure 8, the voltage feedback control strategy was started at 0.2 s, and the harmonic content of each phase voltage at PCC point of the device decreased from about 6.8% to 2%. The harmonic control accuracy was high, but it did not reach the steady state until 0.4 s, and the response time of reaching to the steady state was long. In Figure 9, the current feed-forward control strategy was started at 0.2 s, and the harmonic content of each phase voltage at PCC point of the device decreased from about 6.8% to 4%. The harmonic control accuracy was general, but it reached the steady state at 0.25 s and the response speed was fast. In Figure 10, the composite control strategy of feedback and feed-forward was started at 0.2 s, and the harmonic content of each phase voltage at PCC point of the device decreased from about 6.8% to 1.8%. The harmonic control accuracy was high, and it reached the steady state at 0.3 s and the response speed was fast.

Comparing Figure 8, Figure 9 and Figure 10, the voltage feedback harmonic suppression had the characteristic of high control accuracy, and the current feed-forward harmonic suppression had the characteristic of fast response speed. The composite control strategy of feedback and feed-forward proposed in this paper can have the advantages of feedback control and feed-forward control. Meanwhile, it had a good compensation effect and dynamic performance for harmonic voltage suppression.

In order to make a clearer comparison of compensation effect of the feedback and feed-forward composite control strategy, the voltage harmonic content before and after compensation were extracted in this paper, which is shown in Figure 11.

In Figure 11, the contents before and after the output voltage harmonic compensation were 6.66% and 1.87%, respectively. Therefore, the voltage feedback and current feed-forward composite control strategy proposed in this paper can effectively compensate and suppress the harmonic voltage under nonlinear load.

The comparison of response time and suppression effect under three different harmonic control strategies is shown in Table 2. It is obvious that compared with the single control strategy of voltage feedback and current feed-forward, the harmonic suppression effect under the composite control strategy was better. The response time was only 100 ms, which fully met the control requirements.

### 4.3. Comparison and Analysis of Algorithms under Dynamic Load Changes

In order to verify the tracking and response ability of the proposed control strategy under nonlinear load dynamic changes, it was verified by simulation in this paper, and the results are shown in Figure 12.

In Figure 12, the control strategy started at 0.2 s, the nonlinear load changed at 0.3 s and the harmonic current on the load side increased, resulting in an increase in the harmonic content of the output voltage. Among them, under the voltage feedback control strategy, the fluctuation range of the total harmonic wave content was about 2%, and the time required to reach the steady state again was ≥100 ms. Under the current feed-forward control strategy, the fluctuation range of the total harmonic wave content was about 1.5%, the time required to reach the steady state was only 40 ms. Under the composite control strategy, the fluctuation range of the total harmonic wave content was about 1%, and the time required to reach the steady state again only took 40 ms. By comparison, it can be seen that if the nonlinear load changed dynamically, the composite control strategy proposed in this paper had more competitive advantages in response speed, fluctuation range and harmonic suppression effect than the single harmonic suppression strategy.

## 5. Experimental Test

In order to verify the composite control strategy and simulation test results in this paper, an experimental platform was built according to Figure 7, and the experimental parameters of the platform are shown in Table 3.

The experimental platform controller was the digital signal processor TMS320F28069 and the experimental waveforms were recorded by Fluke Oscilloscope 190–204. Figure 13 shows the experimental waveform of a nonlinear load.

Figure 13 shows the output voltage waveforms of the power quality control device, Figure 12 compares the C-phase voltage harmonic of device before and after adopting the compound control strategy based on feedback and feed-forward proposed in this paper and Figure 13c,d compares the total output voltage harmonic of each phase before and after adopting the compound control strategy. In order to verify the suppression effect of the proposed control strategy on a specific order of harmonics, this paper conducted a comparative test on the compensation effects of the 5th and 7th harmonics of the output voltage of the device. In Figure 13a,b, it can be clearly seen that after adopting the feedback and feed-forward harmonic voltage mixed compensation strategy, the total harmonic of c-phase voltage was reduced from 8.75% to 3.61% the 5th harmonic was reduced from 6.7% to 0.1% and the 7th harmonic was reduced from 5.1% to 0.4%.

## 6. Conclusions

A novel composite control strategy based on voltage feedback and current feed-forward was proposed in this paper, which can solve the problem of high voltage harmonic content at the common connection point of the power quality control device. The simulation and experimental results verified the effectiveness of the compound control strategy. The proposed compound control strategy had the following characteristics:

1. The current feed-forward control in the compound control strategy can reduce the harmonic impedance of the device at each frequency by introducing virtual harmonic impedance, so it can effectively reduce the voltage distortion rate at the common connection point;

2. The proposed composite control strategy can reduce the harmonic voltage content of the common connection point from 6.8% to 1.8%, and the response time was only 100 ms. It met the stringent requirements of the high-end manufacturing industry for high voltage harmonic compensation accuracy and fast response time;

3. Based on the harmonic suppression effect tested and analyzed on the 380 V low-voltage test platform, we believe the control strategy proposed in this paper is also applicable to the cascaded power quality control device at the 10 kV voltage level.

## Figures and Tables

**Figure 1 micromachines-13-01646-f001:**
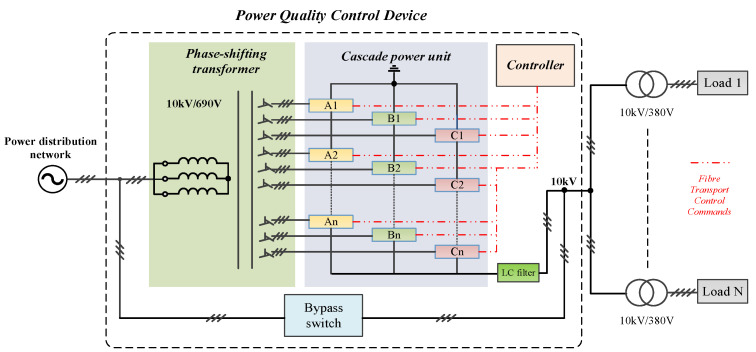
Main loop diagram of the cascaded power quality improvement device.

**Figure 2 micromachines-13-01646-f002:**
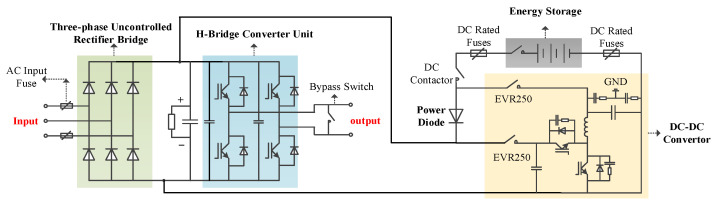
Topology diagram of the cascade power unit.

**Figure 3 micromachines-13-01646-f003:**
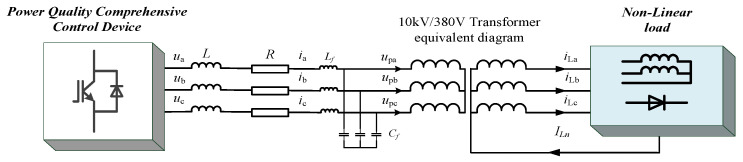
Equivalent model of circuit on the output side of the device.

**Figure 4 micromachines-13-01646-f004:**
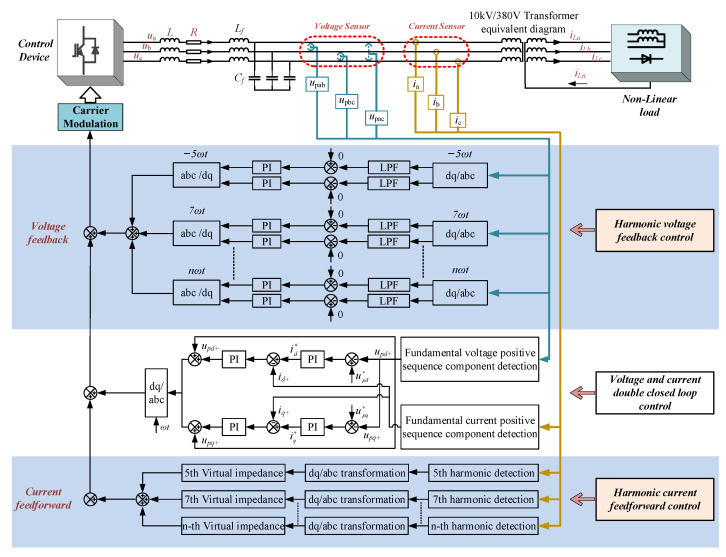
The compound control strategy of voltage feedback + current feed forward.

**Figure 5 micromachines-13-01646-f005:**
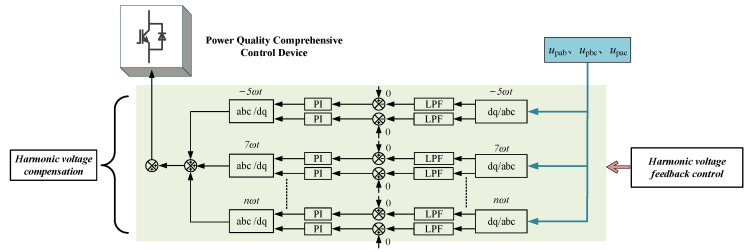
Voltage feedback control block diagram.

**Figure 6 micromachines-13-01646-f006:**
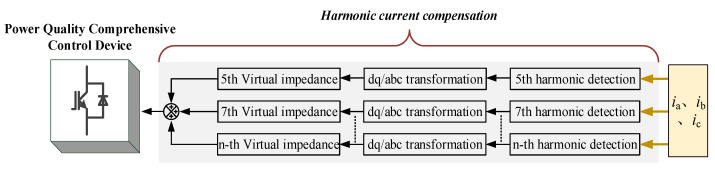
Block diagram of current feed-forward control based on virtual impedance.

**Figure 7 micromachines-13-01646-f007:**
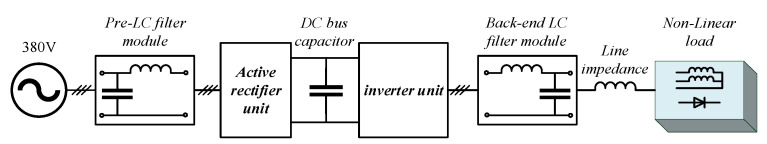
The system simulation model and experimental platform.

**Figure 8 micromachines-13-01646-f008:**
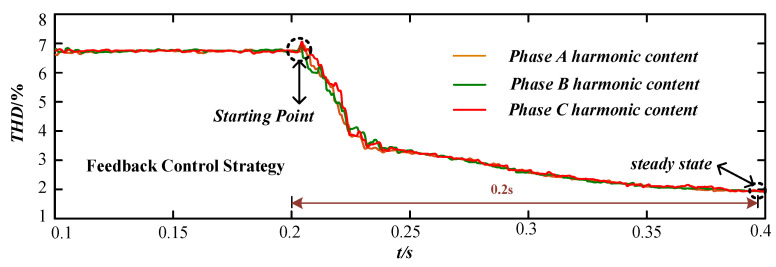
The harmonic change under the voltage feedback control strategy.

**Figure 9 micromachines-13-01646-f009:**
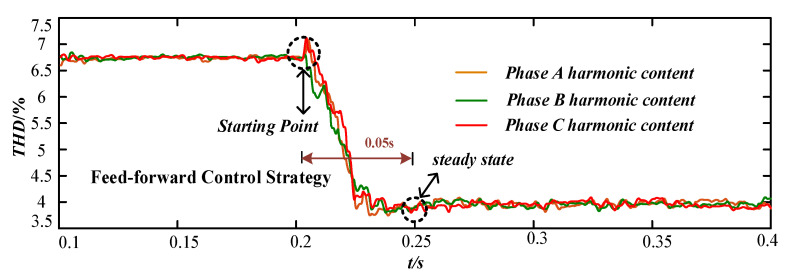
The harmonic change under the current feed-forward control strategy.

**Figure 10 micromachines-13-01646-f010:**
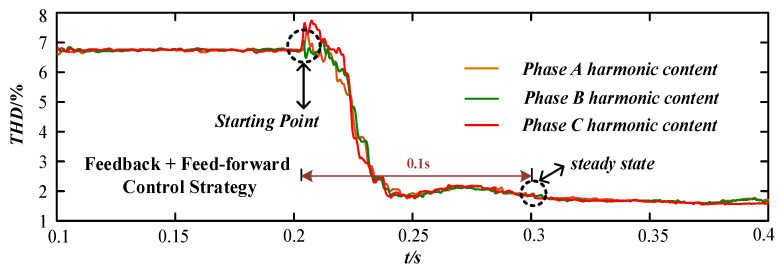
The harmonic change under the compound control strategy of feedback + feed-forward.

**Figure 11 micromachines-13-01646-f011:**
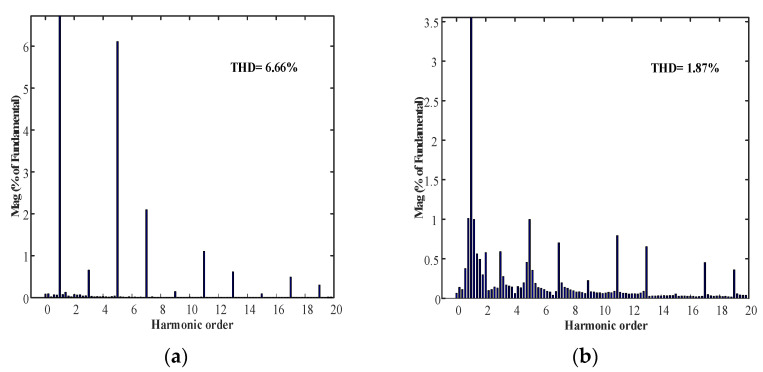
Comparison of before and after voltage harmonic content compensation: (**a**) harmonic content before compensation; (**b**) harmonic content after compensation.

**Figure 12 micromachines-13-01646-f012:**
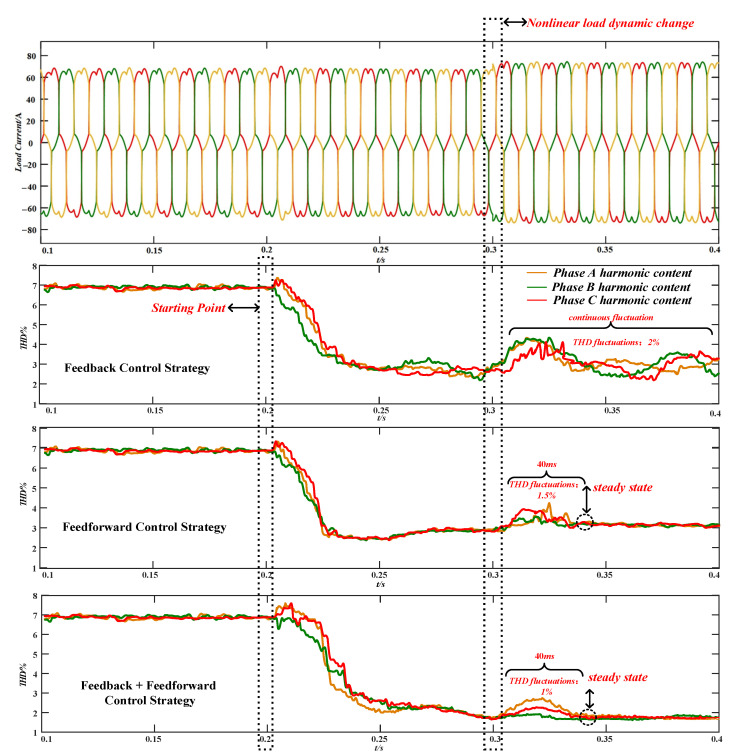
Harmonic suppression effects of different control algorithms under nonlinear load dynamic changes.

**Figure 13 micromachines-13-01646-f013:**
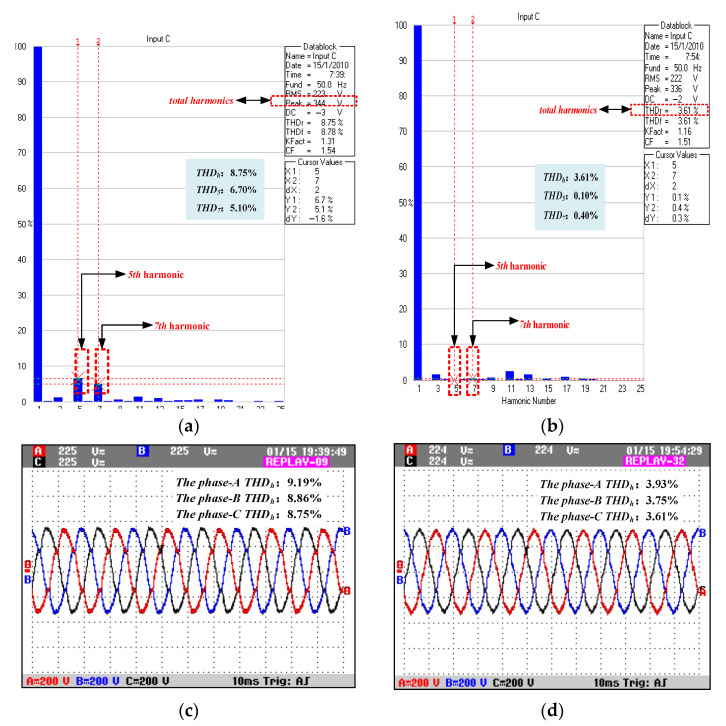
Output voltage waveforms of the nonlinear load: (**a**) C-phase harmonic percentage without compensation; (**b**) C-phase harmonic percentage with compensation; (**c**) voltage waveform before compensation; (**d**) voltage waveform after compensation.

**Table 1 micromachines-13-01646-t001:** System parameters.

Parameters	Value
Frequency of 3-phase AC voltage/Hz	50
rms voltage of 3-phase AC voltage/V	380
Voltage of DC capacitor/V	650
DC bus capacitor/μF	5440
Inductance of the front-end LC filter module/μH	500
Capacitor of the front-end LC filter module/μF	200
Inductance of the back-end LC filter/μH	400
Capacitor of the back-end LC filte/μF	200
Line impedance/Ω	0.4
Resistance of nonlinear load/Ω	10

**Table 2 micromachines-13-01646-t002:** Comparison of harmonic suppression effect and response time.

Number	Name	Voltage Feedback Control Strategy	Current Feed-Forward Control Strategy	Compound Control Strategy
1	Start/ms	200	200	200
2	Steady/ms	400	250	300
3	Response Time/ms	200	50	100
4	Before Harmonic Suppression/%	6.8	6.8	6.8
5	After Harmonic Suppression/%	2	4	1.8

**Table 3 micromachines-13-01646-t003:** System parameters of the experimental platform.

Parameters	Value
Three-phase AC voltage frequency/Hz	50
Three-phase AC voltage RMS/V	380
Switching frequency/kHz	12
DC bus voltage/V	650
DC bus filter capacitor/μF	5440
Input side filter inductance/μH	300
Input side filter capacitor/μF	30
Output side filter inductance/μH	250
Output side filter inductance/μF	30
Line impedance/Ω	0.7
Nonlinear load resistance/Ω	30

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
