# Peer review of "Investigation of an Output Voltage Harmonic Suppression Strategy of a Power Quality Control Device for the High-End Manufacturing Industry"

_micromachines, 2022, doi:10.3390/mi13101646_

Round 1

Reviewer 1 Report

In my opinion the paper is well structured and the mhetod to impove power quality proposed is getting good results. I am verry curios about how it will work on 10 kV.

Author Response

Dear Editor and Reviewers,

The authors would like to thank the reviewers for their encouraging and insightful comments, and appreciate your useful and significant comments and input pushing the work to be improved.

Please see the attachment for the responses to the review comments.

Finally, Thanks for the reviewer's recognition of our work and the kind suggestions.

Reviewer 2 Report

The paper presents a composite control strategy based on voltage feedback and current feed-forward power quality control device to solve the problem of high voltage harmonic content at the PCC. The paper is well written and organized, However, I have the following comments:

1-     Comparative analysis is required with literature to highlight the contribution. At the same time, the literature review should be improved to support the objectives.

2-     The paper has not reported its outcome with respect to failure due to occurrence of sudden faults. Issue needs to be addressed.

3-     There are many variables, so nomenclature list at the beginning of the text will be useful.

4-     How many sensors you need for your control? Could it be limiting factor regarding robustness, reliability and costs? Some comments are needed in the text.

5-     Simulation Results should be enriched. More detailed scenarios should be added.

6-     In general, what makes this method superior to many others previously reported? Is just a supposedly lower need of computation?

Author Response

Dear Editor and Reviewers,

The authors would like to thank the reviewers for your encouraging and insightful comments, and appreciate your useful and significant comments and input pushing the work to be improved.

The manuscript has been revised, considering all the comments from the reviewers and editor. The authors have addressed the comments with all necessary modifications and responses. The salient additions to the paper were highlighted in red. In addition, the detailed clarifications to respective reviewers are listed as follows.

Finally, Thanks for the reviewer's recognition of our work and the kind suggestions.

Round 2

Reviewer 2 Report

The paper is well revised; All the concerns in my previous report have been well addressed.